# Development of [^225^Ac]Ac-PSMA-I&T for Targeted Alpha Therapy According to GMP Guidelines for Treatment of mCRPC

**DOI:** 10.3390/pharmaceutics13050715

**Published:** 2021-05-13

**Authors:** Eline L. Hooijman, Yozlem Chalashkan, Sui Wai Ling, Figen F. Kahyargil, Marcel Segbers, Frank Bruchertseifer, Alfred Morgenstern, Yann Seimbille, Stijn L. W. Koolen, Tessa Brabander, Erik de Blois

**Affiliations:** 1Erasmus Medical Centre, Department of Radiology and Nuclear Medicine, 3015 CN Rotterdam, The Netherlands; e.hooijman@erasmusmc.nl (E.L.H.); y.chalashkan@erasmusmc.nl (Y.C.); s.ling@erasmusmc.nl (S.W.L.); f.kahyargil@erasmusmc.nl (F.F.K.); m.segbers@erasmusmc.nl (M.S.); y.seimbille@erasmusmc.nl (Y.S.); s.koolen@erasmusmc.nl (S.L.W.K.); t.brabander@erasmusmc.nl (T.B.); 2Erasmus Medical Centre, Department of Pharmacy, 3015 CN Rotterdam, The Netherlands; 3Joint Research Centre, European Commission, 76344 Karlsruhe, Germany; frank.BRUCHERTSEIFER@ec.europa.eu (F.B.); Alfred.MORGENSTERN@ec.europa.eu (A.M.); 4Division of Life Sciences, TRIUMF, Vancouver, BC V6T 2A3, Canada; 5Department of Medical Oncology, Erasmus MC Cancer Institute, 3015 CN Rotterdam, The Netherlands

**Keywords:** actinium-225, [^225^Ac]Ac-PSMA-I&T, clinical translation, good manufacturing practice (GMP), metastatic castration-resistant prostate cancer (mCRPC), prostate-specific membrane antigen (PSMA) therapy and imaging (I&T), targeted alpha therapy (TAT)

## Abstract

Recently, promising results of the antitumor effects were observed in patients with metastatic castration-resistant prostate cancer treated with ^177^Lu-labeled PSMA-ligands. Radionuclide therapy efficacy may even be improved by using the alpha emitter Ac-225. Higher efficacy is claimed due to high linear energy transfer specifically towards PSMA positive cells, causing more double-strand breaks. This study aims to manufacture [^225^Ac]Ac-PSMA-I&T according to good manufacturing practice guidelines for the translation of [^225^Ac]Ac-PSMA-I&T into a clinical phase 1 dose escalation study. Quencher addition during labeling was investigated. Quality control of [^225^Ac]Ac-PSMA-I&T was based on measurement of Fr-221 (218 keV), in equilibrium with Ac-225 in approximately six half-lives of Fr-221 (T½ = 4.8 min). Radio-(i)TLC methods were utilized for identification of the different radiochemical forms, gamma counter for concentration determination, and HPGe-detector for the detection of the radiochemical yield. Radiochemical purity was determined by HPLC. The final patient dose was prepared and diluted with an optimized concentration of quenchers as during labeling, with an activity of 8–12 MBq (±5%), pH > 5.5, 100 ± 20 μg/dose, PSMA-I&T, radiochemical yield >95%, radiochemical purity >90% (up to 3 h), endotoxin levels of <5 EU/mL, osmolarity of 2100 mOsmol, and is produced according to current guidelines. The start of the phase I dose escalation study is planned in the near future.

## 1. Introduction

Currently, prostate cancer is one of the lead diagnoses of cancer in men. Worldwide, 1.2 million cases of prostate cancer were estimated in 2018. This number is expected to increase as the population grows and life expectancy increases [1]. Approximately 10–20% of the estimated cases develop into castration-resistant prostate cancer within 5 years of diagnosis, and more than 80% develop into metastatic castration-resistant prostate cancer (mCRPC) [2]. Nowadays, mCRPC patients are treated with hormone therapy and chemotherapy, which can have severe side effects. Prostate cancer is considered castration-resistant when the disease progression continues; even after androgen deprivation therapy is initiated [3]. Recently, interests have grown in the field of nuclear medicine towards a theranostic for the treatment of mCRPC.

The prostate-specific membrane antigen (PSMA) is a type II transmembrane glycoprotein with a domain both intracellular and extracellular and is highly upregulated in 90–100% of prostate cancer cases. High PSMA expression is seen in the further progression of the disease [4]. Although PSMA is also expressed on the benign prostate epithelium and on other tissues, such as the kidneys, small intestine, and the salivary glands, the expression on prostate cancer cells is a thousand-fold higher than the expression on normal tissues. Therefore, PSMA has been identified as a promising target for both imaging and therapy of prostate cancer (PCa) [5]. PSMA-ligands with a DOTA- and DOTAGA-chelator (such as PSMA-617 and PSMA-I&T) have been developed to label with both diagnostic and therapeutic radionuclides. For Positron Emission Tomography (PET) imaging, PSMA-ligands can be labeled with radionuclides, such as Gallium-68 (Ga-68) and Fluorine-18 (F-18). For radionuclide therapy, PSMA-ligands can be labeled with radionuclides, such as Lutetium-177 (Lu-177) and Actinium-225 (Ac-225) [6].

Several studies using [^177^Lu]Lu-PSMA-617 and [^177^Lu]Lu-PSMA-I&T as a therapy have shown to be very effective in treating PCa [7,8,9,10,11,12,13,14,15,16], while at the same time being well tolerated, with minor side effects. It is expected that an alpha-emitter would be even more effective than a beta-emitter, such as [^177^Lu]Lu-PSMA. The use of alpha-emission offers advantages over beta-emission due to the high linear energy transfer (LET) and the limited range in tissue [17]. The high LET effectively kills tumor cells through DNA double-strand and DNA cluster breaks, and the limited range allows selective tumor cell killing while sparing healthy tissue [18,19]. In the past few years, the alpha-emitting radionuclide Ac-225 has been subject to several investigations as a potential radionuclide in targeted alpha therapy (TAT). Ac-225 is an alpha emitter with a half-life of 9.92 days, which is an appropriate half-life for convenient treatment. Decay of Ac-225 occurs via a cascade ending into stable Bismuth-209 (Bi-209) via six short-lived daughter radionuclides and emission of four alpha particles. Total energies of alpha-particles range from 5.8 to 8.4 MeV with tissue ranges of 47 to 85 µm. The cascade also includes two beta-disintegrations of 1.6 and 0.6 MeV, and minor gamma co-emissions are generated from disintegrations of Fransium-221 (Fr-221) and Bismuth-213 (Bi-213), which can be used for detection [20].

Promising results were shown by preclinical in vivo and in vitro studies, as well as first reports on the use of Ac-225-labeled PSMA in the clinic [4]. Previous preliminary clinical studies using [^225^Ac]Ac-PSMA-617 and [^225^Ac]Ac-PSMA-I&T showed antitumor effects, and the treatment was generally well-tolerated [21,22,23,24]. Treatment with [^225^Ac]Ac-PSMA-617 resulted in a significant PSA reduction in patients diagnosed with mCRPC and had a promising outlook [24,25,26]. Even after limited efficacy of [^177^Lu]Lu-PSMA therapy, [^225^Ac]Ac-PSMA shows substantial antitumor effects [27]. However, [^225^Ac]Ac-PSMA is not yet evaluated on safety, tolerability, biochemical effects, and dosimetry in a formal phase 1 study. In the past, the labeling was assessed only with the radio-(i)TLC-scanner. However, in this study, additional measurements are performed with the HPGe-detector and gamma counter. Additionally, the stability is monitored by HPLC for the first time. The objective of our study is to develop an [^225^Ac]Ac-PSMA-I&T production method, including methods for quality control according to the GMP regulations to support a phase I dose escalation study.

## 2. Materials and Methods

### 2.1. Materials and Chemicals

The preparations that are described for the radiolabeling of PSMA-I&T with Ac-225 were performed in a dedicated glovebox (class A). All chemicals used for labeling and quality control were in accordance with the European Pharmacopoeia (Ph. Eur.) regulations unless stated otherwise. Ac-225 was diluted into 0.1 M HCl (Eckert & Ziegler, Berlin, Germany). Stock solutions (10 mL) were prepared in quartz-coated sterile vials (Curium, Petten, The Netherlands). All purchased chemicals were prepared with Milliq water (HiPerSolv Chromanorm, VWR Chemicals, Amsterdam, The Netherlands). Stock solutions prepared the day before labeling were 1 M HCl (from 37% HCl), 10 M NaOH (Sigma Aldrich Chemicals BV, Zwijndrecht, The Netherlands) and 0.1 M TRIS-buffer pH 9 (Merck chemicals BV, Amsterdam, The Netherlands). Two stock solutions were prepared on the day of labeling: First, 20% ascorbic acid (VWR Chemicals BV, Amsterdam, The Netherlands) was prepared; the ascorbic acid solution was transformed to ascorbate by addition of 10 M NaOH to a pH 5.8. Secondly, PSMA-I&T (250 µg, Huayi Isotopes Co. via ATT Scintomics, Fürstenfeldbruck, Germany) was dissolved in 0.1 M TRIS-buffer (pH 9) to a concentration of 600 µg/mL. Directly after labeling, 4 mg/mL diethylenetriaminepentaacetic acid (DTPA) (Hospital pharmacy A15, Gorinchem, The Netherlands) was added to the labeling mixture. A solution for injection was prepared by the addition of ascorbate (50% *v*/*v*) and ethanol (6% *v*/*v*, 96%) (Hospital pharmacy A15, Gorinchem, The Netherlands) into saline.

The Ac-225 was provided by the Joint Research Centre (JRC, Karlsruhe, Germany) and is a radionuclide that is derived from a Th-229 source eluted by JRC. Ac-225 detection was performed via Fr-221 and was based on the 218 keV gamma emission (yield of ± 11%). After six half-lives of Fr-221 (T½ = 4.8 min, ~30 min), a secular equilibrium was formed with Ac-225 (Figure 1) [21]. After 30 min, the amount of Fr-221 activity was equal to 98.5% of the activity of the Ac-225. Therefore, 30 min after applied separation, the gamma rays were thus proportional to the amount of activity of Ac-225 in the sample, which was much faster in comparison to the Ac-225–Bi-213 equilibrium (~4.5 h). Practically, it means that ITLC-strips or collected fractions from (indirect) measurements, such as radio-(i)TLC and HPLC, need to be measured after 30 min of separation of Ac-225 from its daughters.

The radiolabeling was performed in a V-shaped Biotage vial (0.5–2.0 mL). Vials with a rubber stopper were cleaned under class A by soaking the vials in 10% ethanol, Milliq water, 1 M HCl, and Milliq water for an hour, respectively, to avoid any metal-ion contamination. After drying, the vials were packaged and sterilized by gamma radiation with a dose of 25 kGy (Steris, Ede, The Netherlands).

Standardized geometries were used for radioactivity analysis (Table 1). The radio-(i)TLC-strips with standardized size were used; the activity was spotted at a retention factor (rf) of 1 cm from the bottom on the standardized strips. To prevent contaminations with activity, after separation was performed, the ITLC-strips were sealed into parafilm. The strips were measured in the HPGe-detector accordingly, in a borosilicate glass tube, which was placed into a 10 mL reaction tube. For the activity concentration measurements, <10 kBq of Ac-225 was prepared in a total volume of 1 mL solution in the same geometry.

For radio-(i)TLC, two eluents were used. (1) Sodium citrate (J.T. Baker, Philipsburg, MT, USA) was prepared at a concentration of 0.5 M and pH 5. (2) Acetonitrile (Honeywell) was prepared in Milliq water at a 1:1 (*v*/*v*) ratio. During the validation phase, for identification of free Ac-225 and [^225^Ac]Ac-DTPA, a solution of Ac-225 (10 kBq/µL) and a solution of [^225^Ac]Ac-DTPA (10 kBq/µL) were prepared.

Two different solvents were used to perform HPLC: (A) Water containing 0.1% TFA and 5% acetonitrile, and (B) acetonitrile containing 0.1% TFA and 5% water. All HPLC samples were prepared in the glovebox. For injection, a dilution from the labeling mixture was made to a concentration of ~10 kBq/100 µL into a 300 µL polypropylene vial (Waters, Etten-Leur, Netherlands). SPE Sep-Pak C18 purification followed by HPLC analysis was performed to confirm that free Ac-225, [^225^Ac]Ac-DTPA, and [^225^Ac]Ac-PSMA-I&T can be monitored separately with the HPLC. The labeling mixture containing [^225^Ac]Ac-PSMA-I&T was added into the Sep-Pak C18 and was eluted with 3 mL water and 3 mL ethanol, respectively. Both eluents were measured in the dose calibrator, and the ethanol fraction was injected into the HPLC.

### 2.2. Radiolabeling

For the radiolabeling, 30 MBq of Ac-225 was dissolved in a V-vial by the addition of 30 µL of 0.1 M HCl. The stock was used after incubation of >30 min. The activity was measured by a calibrated dose calibrator, and the measurement was corrected for geometry. 300 µL of 300 or 600 µg/mL PSMA-I&T was used during labeling. PSMA-I&T (0.1 M TRIS buffer (pH 9)) was added into a Biotage microwave vial. Then, 80 µL of Milliq water and 200 µL of ascorbate solution (1 M, pH 5.8) was added. An activity of 15, 17, or 19 MBq (patient dose-dependent) of Ac-225 was added and corrected to a volume of ~25 µL of 0.1 M HCl. The final labeling volume (600 µL) was kept constant using Milliq water. The PSMA-I&T was labeled with Ac-225 by heating the solution for 6 min at 95 °C in the Biotage microwave (Figure 2).

After labeling, 500 µL of ascorbate buffer (1 M, pH 5.8) was added for stabilization. Furthermore, 15 µL of DTPA (4 mg/mL) was added to capture the remaining Ac-225 and daughters [29,30,31]. A dose calibrator was used for intermediate radioactivity measurements. The amount of ascorbate added during heating was investigated (0, 50, 100, 200 µL) to optimize the stability of [^225^Ac]Ac-PSMA-I&T. The stability of [^225^Ac]Ac-PSMA-I&T was evaluated by injecting samples into HPLC, which were collected in fractions, and the fractions were measured with a gamma counter. Optimal reaction conditions were used to perform further radiolabeling of PSMA-I&T (Table 2).

### 2.3. Systems and Settings

The validation of the used equipment (dose calibrator, radio-(i)TLC, HPGe-detector, gamma counter, and HPLC) and related quality control techniques were based upon the European Association of Nuclear Medicine (EANM) guidelines [32] and in accordance with Good Manufacturing Practice (GMP) guidelines [33]. From the output of the analytical techniques (radio-(i)TLC and HPGe-detector), the radiochemical yield (RCY) was calculated, which is defined as the ratio (%) between labeled [^225^Ac]Ac-PSMA-I&T and free Ac-225 and/or [^225^Ac]Ac-DTPA (total Ac-225 activity). Furthermore, the radiochemical purity (RCP) was measured by HPLC and is defined as the ratio between intact [^225^Ac]Ac-PSMA-I&T versus other radioactive components present, including radiolyzed [^225^Ac]Ac-PSMA-I&T [34]. The stability is defined as the RCP measured over time. A dedicated cleanroom was equipped and used for preparations of [^225^Ac]Ac-PSMA-I&T. The measurements were kept consistent and were corrected for dedicated geometries during experimental setup, as described (Table 1).

Due to health physics regulations, a specific microwave (Biotage Initiator+, Biotage, Uppsala, Sweden) was used for labeling. The microwave system was placed into a laminar flow cabinet (class C) in the cleanroom. Only Biotage vials could be used since these are certified up to 30 bars, providing additional security. For the radiolabeling, a closed vial containing the labeling mixture was transferred from the glovebox to the microwave and heated to 95 °C for 6 min (effective heating time 5 min). After heating, the vial was cooled to room temperature by a 3-bar nitrogen airflow (<1 min).

A dose calibrator (VIK-202, Comecer, Castel Bolognese, Italy) was used for intermediate activity measurements. The settings were adapted from the Tc-99m channel, calculating an adaptive factor (factor = 148), and measurements were performed after 30 min. For the preparation of the final patient dose, a syringe was filled with 3–8 mL from filtered labeling stock with a final activity of 8, 10, or 12 MBq. To cross-calibrate the dose calibrator, measurement of a 5 µL sample from the same filtered labeling stock was measured on the gamma counter (<10 kBq/µL), as described later.

The radio-(i)TLC-Scanner (bSCAN, Brightspec, Zelik, Belgium) was placed in a fume hood and was equipped with a NaI(Tl) scintillator detector, 2.54 × 2.54 cm crystal size, and digital multichannel analyzer (MCA). Measurements with different speeds up to 0.60 cm/min (max 1000 s) from the bottom to the top of the strip were evaluated. Linearity was measured for activities ranging from 1 to 100 kBq of Ac-225, with a limit of detection (LOD) of <1 kBq). To confirm optimal separation of the radiochemical species, for the determination of the RCY of [^225^Ac]Ac-PSMA-I&T, 3 ITLC-strips were prepared for each eluent. During the validation of the quality control, 5 µL of the labeling solution (±50 kBq) was deposited on each strip. After drying, the strips were placed in the mobile phases; sodium citrate (0.5 M, pH 5) and acetonitrile/Milliq water (1:1 (*v*/*v*)). When the front reached the top, the strips were dried and left for 30 min and 24 h, respectively.

A GC1018 High Purity Germanium (HPGe, Miron Technologies Canberra, Olen, Belgium) detector was located in a dedicated quality control lab. The HPGe-detector has a high-energy resolution (<1.2 keV @ 122 keV and <1.8 keV @ 1332 keV) and is, therefore, highly suitable for the measurement of Fr-221 and Bi-213 separately. Downscatter corrections were standardly applied in the software, permitting independent measurement of Fr-221 (218 keV) and Bi-213 (440 keV). After running 3 additional ITLC-strips as described previously, strips were cut into two parts to 0 < Rf < 0.7 (bottom) and Rf > 0.7 (top). The rf regions were proven to be accurate and were based upon the validated radio-(i)TLC-scanner data. Activity measurements of Fr-221 were evaluated for 1 min/sample and counts of dedicated peak areas were used for RCY calculations. The RCY was obtained by calculating the relative ratio between the bottom and the total activity. The RCY could be measured with an accuracy of ~1%.

The automated gamma counter 2480 Wizard-2 (Perkin Elmer, Waltham, MA, USA) was used for Fr-221 measurements. The detector system (thallium activated, sodium iodide crystal) was calibrated for the Fr-221 energy window (186–226 keV). A calibration factor was determined by using a reference standard supplied by JRC Karlsruhe. A small (5 µL) sample of the filtrate was measured (<10 kBq/1 mL) in the standardized geometry on Fr-221 (1 min/sample) to determine the activity concentration of the final patient dose. The measurements were performed with a precision of ~1% and an accuracy of ~2%. The gamma counter has shown to be 15 times more sensitive than the radio-(i)TLC-scanner and, therefore, is highly accurate for concentration determination.

An Alliance 2695XE HPLC (Waters Chromatography B.V., Etten-Leur, The Netherlands) including a PDA detector (W2298) containing a C18 column (LiChrospher RP-18 endcapped (5 µm)) Merck, Amsterdam, The Netherlands) with dedicated software (Empower 3) was used to determine RCP and stability. The gradient used: 0–2 min 100% solvent A, 2–10 min to 100% B, 10–15 100% solvent B, 15–20 100% solvent A, with a flow of 1 mL/min. For RCP measurements, a sample of 100 µL/10 kBq [^225^Ac]Ac-PSMA-I&T was injected into the HPLC. The HPLC eluate was collected by an autosampler (15 drops/tube) during the gradient (20 min).

### 2.4. Quality Control

The release of the final batch of [^225^Ac]Ac-PSMA-I&T depends on the amount of activity, RCY, RCP, and sterility tests. Quality controls were validated according to EANM guidelines in the previously described geometries (Table 1). Intermediate activity measurements were performed with the dose calibrator before and after labeling. For the release of the final patient dose, a correction factor was used, for the exact type of syringe containing the final [^225^Ac]Ac-PSMA-I&T solution. For the radio-(i)TLC, different mobile phases were investigated (as described in Section 2.1). With sodium citrate, the labeled product stayed at the origin, and with acetonitrile/water, the labeled product migrated toward the top of the strip when separated. For each eluent, a sample was measured with the radio-(i)TLC-scanner using optimized settings (Fr-221 window, 900 s). Furthermore, three ITLC-strips were measured with the HPGe-detector. The RCY measured by the radio-(i)TLC-scanner, and the three ITLC-strips measured by the HPGe-detector must have a RCY of >95% and are part of the release criteria of [^225^Ac]Ac-PSMA-I&T. The RCP was evaluated by HPLC time-dependently. The HPLC fractions were measured >30 min after collection in the gamma counter at 0, 1, and 3 h (*n* = 3) after labeling. Measured counts from the fractions (Fr-221) were plotted in accordance with their tube number to obtain a HPLC chromatogram. A RCP of >90% is considered within the release criteria.

### 2.5. Preparation Patient Solution

The solution for injection was prepared by dissolving the requested activity (8–12 MBq) into an injection solution of ascorbate (final concentration of ~0.5 M [35,36]), ethanol (final concentration of 5%), and saline, which resulted in a total injection volume of 3–8 mL. Sterile filtration (0.22 µM) of the patient dose was performed, and the integrity of the filter was checked by performing a bubble point test. From a small sample of the filtrate (0.2 mL), endotoxin testing was performed, as well as determination of the activity concentration. The activity concentration, with the standardized geometry, was analyzed in the gamma counter.

## 3. Results

After validation and calibration of the systems (dose calibrator, radio-(i)TLC, HPGe-detector, gamma counter, and HPLC) for Ac-225 (Fr-221 equilibrium), quality control techniques were applied. Radio-(i)TLC-analysis was evaluated, the final scanning settings were 0.66 cm/min (900 s), 5 µL/50 kBq to obtain ~20.000 cps (<1% error). Final radio-(i)TLC-scan parameters were: radiopharmaceutical (rf < 0–0.7), background (rf < 0.1), and impurity (rf > 0.7–1.0). Two radio-(i)TLC mobile phases were tested to evaluate the optimal conditions to identify Ac-225 and [^225^Ac]Ac-DTPA. The sodium citrate mobile phase resulted in a good separation between the free Ac-225, [^225^Ac]Ac-DTPA, and [^225^Ac]Ac-PSMA-I&T, even when a poorly labeled batch (RCY < 50%) was analyzed by this method (Figure 3a). An inadequate separation was observed with acetonitrile/water (1:1 *v*/*v*) (Figure 3b). Ac-225 and [^225^Ac]Ac-DTPA were also individually investigated in additional analysis, and both were confirmed at rf > 0.7, data not shown. No separation could be performed between radiolyzed [^225^Ac]Ac-PSMA-I&T and intact [^225^Ac]Ac-PSMA-I&T on the radio-(i)TLC-scanner.

The first labeling during the development phase already showed a RCY of >95%. Unfortunately, the labeling solution did not reach the preset quality specifications regarding stability (RCP > 90%) (Figure 4a). A peak could be seen in fraction 4–9, as well as in fraction 10–15, which resulted in a RCP of <84% (T = 0 h) (Table 3). A SPE Sep-pak purification was performed from the labeling mixture, separating the free Ac-225 and [^225^Ac]Ac-DTPA from the radiolyzed PSMA-I&T and [^225^Ac]Ac-PSMA-I&T. The ethanol eluent was injected into the HPLC and thereafter analyzed with the gamma counter. After SPE Sep-Pak purification, impurities (fractions 4–9 and fractions 10–15) remained present, which resulted in a similar RCP of ~85% (data not shown). Radiolyzed [^225^Ac]Ac-PSMA-I&T remained present in the peak (fractions 4–9) and as a smear (fractions 10–15) in between the first peak and the [^225^Ac]Ac-PSMA-I&T. A [^225^Ac]Ac-PSMA-I&T batch with a RCY ~35% was injected into the HPLC for further verification of impurity detection. This resulted in the HPLC chromatogram as shown in Figure 4b. The free Ac-225 and [^225^Ac]Ac-DTPA (65%) were detected as overlying peaks on HPLC chromatogram and were both eluted within the dead volume of the HPLC system (retention time (rt) ~3 min, fraction 4–9).

As described previously, the increased number of counts in fractions 4–15 was correlated to radiolyzed [^225^Ac]Ac-PSMA-I&T. The addition of 1 M ascorbate as a quencher was investigated to increase the purity of the [^225^Ac]Ac-PSMA-I&T. Different amounts of ascorbate addition were tested (0, 50, 100, and 200 µL) during labeling, as well as an increased PSMA-I&T mass (90 and 180 µg). The peaks (fractions 4–15) remained present after the addition of 50 µL, which resulted in a RCP of 93.5% (T = 0 h) to 88.5% (T = 3 h), see Table 3 (>90%, **green**). When 100 µL was added, the RCP after labeling was 93.1% at T = 0 h and 91.5% for T = 3 h, leading to almost an entire reduction in the smear (fractions 10–15).

However, after dilution into saline and filtration, a decrease in stability was observed, below the release criteria (RCP < 90%). An injection solution was prepared containing the same concentration of quencher (ascorbate) as during labeling to increase the stability after dilution and filtration. Additionally, another quencher (ethanol) was introduced into the injection solution [37]. Furthermore, as demonstrated before, the addition of extra ascorbate and an increased PSMA-I&T mass during the labeling step reduced the formation of radiolysis (fractions 4–15). However, this was not enough to reach stability for three hours after dilution into the injection solution and subsequent filtration. Therefore, the amount of ascorbate during labeling was increased to 200 µL (Table 4), resulting in a RCP >90%, which could be maintained up to 3 h after dilution and filtration (*n* = 3, **green**).

Quality control was performed (*n* = 3) after optimization of the dilution and filtration conditions. Preliminary analysis was performed with the radio-(i)TLC-scanner for peak identification and RCY measurements for each batch. As demonstrated by the radio-(i)TLC-scanner analysis, a RCY >95% was obtained (*n* = 6, Figure 5a). The RCY results from the radio-(i)TLC-scanner were similar to the RCY results obtained by the HPGe-detector (RCY > 95%, *n* > 10). The final conditions of labeling, including 200 µL ascorbic acid, 180 µg PSMA-I&T during labeling, and the addition of injection solution for dilution and filtration, resulted in a RCP of >90% (Figure 5b) in the final injection solution, up to 3 h (*n* = 3).

The final patient dose was prepared with an activity of 8–12 MBq (±5%). Furthermore, different parameters were measured and conformed to the release criteria (*n* = 3); pH was determined at >5.5, the PSMA-I&T content was 100 (±20) μg/dose, the RCY >95% for radio-(i)TLC and HPGe-detector measurements, the RCP > 90%, endotoxin levels of <5 EU/mL (*n* > 10) and osmolarity of 2100 mOsmol. Thus, [^225^Ac]Ac-PSMA-I&T was produced within the release criteria and, according to GMP guidelines, as a sterile aqueous solution. The final measurements of the quality control (optimized process) for different batches are summarized in Table 5 (within release criteria, **green**). The patient dose will be infused simultaneously with a saline solution in the ratio of 1:4 to compensate for the high osmolarity.

## 4. Discussion

This study describes the successful production of a 3–8 mL 8–12 MBq [^225^Ac]Ac-PSMA-I&T solution for injection for clinical implementation. To our knowledge, this is the first study reporting the production of [^225^Ac]Ac-PSMA-I&T according to GMP guidelines, annex 13 [38] and 15 [39], in an alpha-lab cleanroom in accordance with the European Pharmacopoeia (Ph. Eur.) [40]. During the development phase, different challenges were faced regarding the radiolabeling and quality control analysis.

The JRC is currently one of few suppliers of Ac-225 for research purposes [17,31]. As published by the EANM in 2008 [41,42], a radioactive precursor for the active pharmaceutical ingredient (API) is allowed in early phase clinical trials when the isotope is not further purified and when the production and quality are monitored closely. Thus, radionuclide purity has to be tested, and long-lived radionuclides (Th-229 and Ra-225) have to be quantified. After the decay of Ac-225, only minor amounts (~5 ppm) of Th-229 and Ra-225 were detected, which are in line with the specifications as provided by the JRC.

As described previously, Ac-225 emits alpha-particles, which are difficult to detect. In order to obtain an accurate measurement of the Ac-225 dose, alternatively, Fr-221 was measured. Fr-221 is in equilibrium with Ac-225 if no chemical separations are involved, and this equilibrium is fundamental for quality control measurements. Furthermore, due to the emission of alpha-particles, Ac-225 is highly toxic [17]. Only low amounts of activity (<10 kBq) were allowed when introducing this highly toxic radionuclide into a clean room environment, due to health physics regulations, for quality control measurements. Detection of <1% impurity is required according to GMP. Therefore, the equipment used to perform the quality control measurements must be very sensitive to be able to detect <100 Bq. For verification of the quality control results of the radio-(i)TLC (RCY), the gamma counter and HPGe-detector can be used. It should be noted that the gamma counter has shown to be more sensitive in comparison to the radio-(i)TLC-scanner. The gamma counter can discriminate more accurately between Fr-221 and Bi-213 than the radio-(i)TLC-scanner. However, the downscatter component of Bi-213 into the Fr-221 energy window depends on the amount of Bi-213. Measurements on the HPGe-detector have a much higher energy resolution (FWHM < 1.5%) compared to the gamma counter (FWHM ~10%). A higher energy resolution enables smaller, more specific energy windows, and therefore, measurements are less biased by downscatter [43]. Moreover, the software of the HPGe-detector corrects for background and remaining downscatter prior to activity calculation. The RCY was, therefore, determined by the HPGe-detector instead of the gamma counter. The gamma counter was still most suitable for activity concentration measurements as it was validated with a calibrated Ac-225 source (provided by JRC). The gamma counter has shown to be highly accurate (~2%) for samples that are in equilibrium.

The stabilization steps demonstrated to be critical during the development process of the labeling of PSMA-I&T. Different aspects of the quenching for [^225^Ac]Ac-PSMA-I&T were of importance for reaching a RCP > 90%. The addition of ascorbate and ethanol was tested to investigate the stability of [^225^Ac]Ac-PSMA-I&T, before and after labeling, due to their quenching characteristics [44,45,46,47]. During labeling, ascorbate was added for stabilization of the labeling solution, ethanol was added in the final injection solution. Ascorbate concentrations (0, 50, 100 µL) were investigated, and the labeled [^225^Ac]Ac-PSMA-I&T before filtration was confirmed to be stable (RCY > 90%, *n* = 3) up to three hours after labeling (addition 100 µL). However, after filtration, the stability decreased (RCY < 87%). A critical step in the production process was the addition of a quencher directly after labeling. Multiple handlings were required before stabilization was established. This created minor deviations among different people. The process was adapted by increasing the amount of ascorbate (1 M) during labeling to 200 µL to avoid any impact on stability during the labeling process. Furthermore, the addition of ethanol in the injection solution (~6% of the final volume) contributed to reaching a final RCP of >90%. Other promising quenchers could be investigated on their efficacy for alpha emitters to increase and maintain a high RCP [45,47].

Another critical aspect to maintaining stability was the PSMA-I&T mass during labeling. This study aimed to administer a standardized PSMA-I&T mass of 100 µg per dose to obtain a comparable biodistribution between patients. Therefore, addition with 90 µg PSMA-I&T before labeling was started. An adjustment of the PSMA-I&T mass was performed after labeling, prior to filtration, to maintain a 100 μg/dose. However, it should be noted that when the PSMA-I&T mass during labeling was increased (180 µg), the RCP also increased (RCP > 90%). This is likely caused by the relatively high number of non-labeled PSMA molecules, which presumably protect the relatively lower number of radiolabeled PSMA molecules present, also expressed as low molar activity.

Direct inline measurement of the alpha emission would be favorable for the HPLC quality control measurements to obtain the RCP. The HPLC was initially combined with a liquid scintillation detector to evaluate inline measurements of Ac-225. However, the direct inline measurement could not be performed since Ac-225 releases four alpha particles, and differentiation between those particles cannot be performed with liquid scintillation. Liquid scintillation could likely be an alternative option for radionuclides emitting a single alpha particle. Additionally, a gamma scintillator detector was also introduced inline. However, Fr-221 activity could not be measured accurately due to the low amounts of activity (<10 kBq) allowed for quality control. Currently, to our knowledge, there are no alternatives for the detection of multiple alpha sources; accordingly, an alternative approach was implemented. After HPLC injection of the stability samples, the HPLC fractions were measured in the gamma counter after 30 min, obtaining indirect measurements. The difference between the sample counts and the background counts introduced statistical variance due to the low activity allowed (<10 kBq).

Furthermore, after the first decay of Ac-225, recoil occurs, leaving free daughter nuclides in the sample [23,48,49,50,51,52]. Analysis of such a sample results in underestimating the true RCP since free daughter nuclides will also be present. Due to the measurement at different time points and thus different incorporation equilibria, an increasing RCP and RCY could be observed in time. To prevent injection of free nuclides into a patient, DTPA was added to complex free Ac-225 and the formed daughters. As described by Miederer et al., 60% of Ac-225 was excreted after injection of [^227^Ac]Ac-DTPA. Thus, the complexation of the free nuclides could be increased by chelator optimization [53,54,55,56], opting for ^225^Ac-H4py4pa, for example [57]. However, further research has to be performed regarding this subject.

In addition, the total time required for the production, including preparations, of the [^225^Ac]Ac-PSMA-I&T was approximately 1.5 days. An important note is that for quality control, the time for release should be taken into account. A limiting factor was that for all quality control measurements, an equilibrium with Fr-221 (>30 min) must have been set before activity measurements can be performed. During the development state, measuring after 30 min supports the preset quality control thresholds, and due to time pressure regarding patient treatment, this time limit could not be increased. In practice, the RCY results increase significantly over time (RCY > 99%) for the radio-(i)TLC, meaning that the actual RCY at 30 min shows an underestimation of the results [58]. The HPLC analysis is not part of the release criteria, as the indirect analysis of the collected fractions of the eluate requires a minimum of 2 h per injection. Hence, HPLC stability measurements were part of the validation (*n* = 3) and will be monitored during the phase I trial.

The final patient dose of [^225^Ac]Ac-PSMA-I&T was prepared with an activity of 8–12 MBq (±5%) and with a high osmolarity of 2100 mOsmol. Therefore, the patient dose will be administered intravenously complementary with a saline solution in the ratio 1:4. As described previously, dilution into saline reduces the stability of the [^225^Ac]Ac-PSMA-I&T solution. Thus, the dilution into saline should be promptly before administration.

Currently, one patient dose can be produced from one batch of obtained Ac-225 at a time due to health physics regulations. However, it would be beneficial to increase the production of the [^225^Ac]Ac-PSMA-I&T by producing multiple patient doses from one batch to improve efficiency and reduce costs.

## 5. Conclusions

In conclusion, a [^225^Ac]Ac-PSMA-I&T 8–12 MBq solution for injection was prepared to confirm the preset release criteria and in accordance with the GMP and Ph. Eur. regulations. The experiments were performed in a pragmatic manner, allowing for reproducibility and thus providing feasibility. Consequently, the implementation of [^225^Ac]Ac-PSMA-I&T solution for injection into the clinic provides a promising outlook for the treatment of mCRPC patients. The start of the phase I dose escalation study is planned in the near future.

## Figures and Tables

**Figure 1 pharmaceutics-13-00715-f001:**
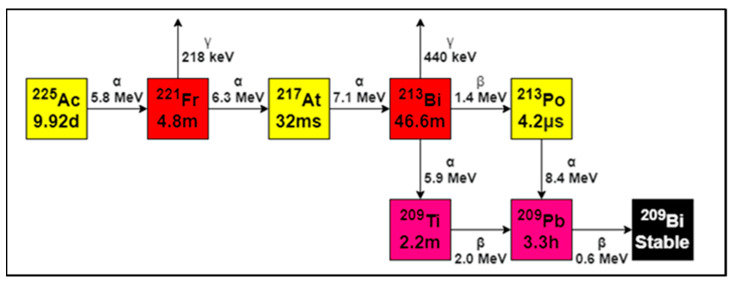
Decay scheme of Ac-225. In total, four alpha-particles are emitted until the stable isotope Bi-209 is formed. Fr-221 and Bi-213 emit gamma radiation (**red**) which was used for detection, 218 keV emission of Fr-221 was in equilibrium with Ac-225. Adapted from [28], Appl. Radiat. Isot., 2013.

**Figure 2 pharmaceutics-13-00715-f002:**
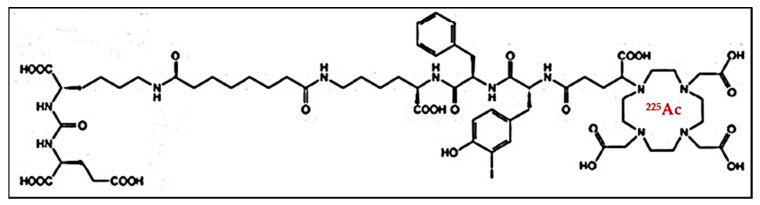
Structural formula of [^225^Ac]Ac-PSMA-I&T, with DOTAGA as a chelator.

**Figure 3 pharmaceutics-13-00715-f003:**
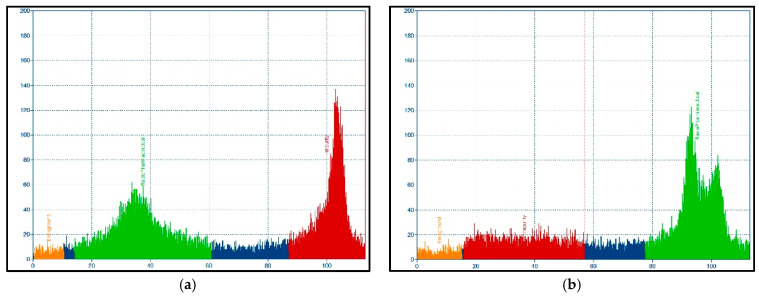
Separation of Ac-225 and [^225^Ac]Ac-DTPA vs. [^225^Ac]Ac-PSMA-I&T with mobile phases sodium citrate (**a**) and acetonitrile/water (**b**). On the x-axis the distance (mm), y-axis present the cps. Radiopharmaceutical ([^225^Ac]Ac-PSMA-I&T, **green**), impurity (Ac-225 and/or [^225^Ac]Ac-DTPA, **red**), background (**orange**), non-selected area (**blue**). Peak separation between reaction mixture was optimal with sodium citrate as mobile phase (RCY ~50%), as confirmed by HPLC analysis.

**Figure 4 pharmaceutics-13-00715-f004:**
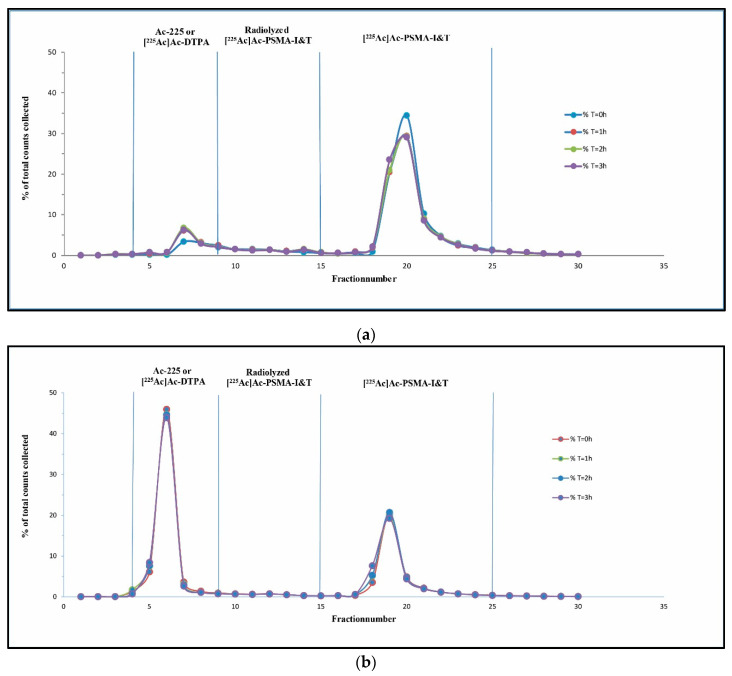
Analysis of HPLC fractions in gamma counter, x-axis represents the tube number of the HPLC fraction (Fr-221 measurements), y-axis % of total counts, measured for 0, 1, 2, and 3 h: HPLC analysis for non-optimized synthesis (RCY < 85%) resulted in a pre-peak (4–9) and a smear that was present in between peaks (10–15) (**a**). Ac-225/[^225^Ac]Ac-DTPA peak at fractions 4–9, for poorly labeled batch (RCY < 35%) (**b**).

**Figure 5 pharmaceutics-13-00715-f005:**
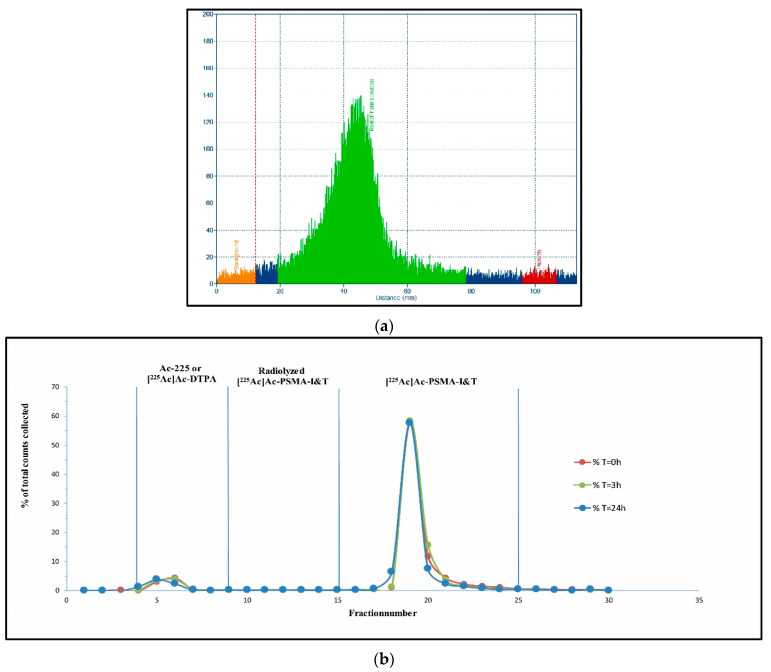
Quality control results for optimal labeling conditions. Radio-(i)TLC (x-axis distance (mm), y-axis cps, measured from bottom to top, radiopharmaceutical (green), impurity (red), background (orange), non-selected area (**blue**)) (**a**) and HPLC chromatogram presenting stability (x-axis represents fraction number, y-axis % of total counts) in time, measured for 0, 3, and 24 h. RCY of >95% and RCP of >90% were maintained up to 3 h (**b**).

**Table 1 pharmaceutics-13-00715-t001:** Standardized geometries for Ac-225/Fr-221 measurements.

**Radio-(i)TLC-scanner**	Strips (ITLC-SG, 0.5 × 10 cm), activity spotted at RF 0.1, sealed with parafilm after separation was performed
**HPGe-detector**	ITLC strip cut at RF 0.7, top and bottom measured in borosilicate glass tube (Pyrex 12 × 7.5 mm) in 10 mL reaction tube (8.5 mL, 75 × 15.7 mm)
**Gamma counter**	Activity concentration of <10 kBq, (5 µL radiolabeling solution) in Milliq water (total volume of 1 mL)
**Dose calibrator**	8–12 MBq, 3–8 mL final solution in 10 mL syringe (Luer Lock) with stopper

**Table 2 pharmaceutics-13-00715-t002:** Optimized reaction conditions of labeling [^225^Ac]Ac-PSMA-I&T.

	Reaction Component	Amount (µL)
**During labeling**	PSMA-I&T (0.1 M TRIS)	300
	Ac-225 + 0.1 M HCl	25 (15, 17, 19 MBq)
	Milliq water	80
	1 M ascorbate	200
**Directly after labeling**	DTPA	15
	1 M ascorbate	500

**Table 3 pharmaceutics-13-00715-t003:** Stability measurements of labeling solution, performed by HPLC injection and analysis by gamma counter.

**No Addition of Ascorbate** **90 µg PSMA-I&T**	**T = 0 h (%)**	**T = 1 h (%)**	**T = 2 h (%)**	**T = 3 h (%)**
Ac-225, [^225^Ac]Ac-DTPA	8.9	13.2	13.3	12.4
Radiolyzed [^225^Ac]Ac-PSMA-I&T	7.5	8.2	8.1	7.6
[^225^Ac]Ac-PSMA-I&T	83.6	78.6	78.6	79.9
**Addition of 50 µL Ascorbate** **90 µg PSMA-I&T**				
Ac-225, [^225^Ac]Ac-DTPA	2.3	4.0	5.2	4.2
Radiolyzed [^225^Ac]Ac-PSMA-I&T	4.3	5.2	6.9	7.3
[^225^Ac]Ac-PSMA-I&T	93.5	90.9	88.0	88.5
**Addition of 100 µL Ascorbate (*n* = 3)** **90 µg PSMA-I&T**				
Ac-225, [^225^Ac]Ac-DTPA	3.1 ± 1.3	3.2 ± 1.5	3.8 ± 1.3	3.5 ± 2.2
Radiolyzed [^225^Ac]Ac-PSMA-I&T	3.9 ± 0.2	4.3 ± 0.6	4.8 ± 1.0	5.2 ± 0.6
[^225^Ac]Ac-PSMA-I&T	93.1 ± 1.1	92.6 ± 1.5	91.6 ± 1.6	91.5 ± 1.6

**Table 4 pharmaceutics-13-00715-t004:** Stability measurement by HPLC injection and gamma counter after dilution and filtration.

Addition of 200 µL Ascorbate and Injection Solution (*n* = 3) 180 µg PSMA-I&T			
	T = 0 h (%)	T = 3 h (%)	T = 24 h (%)
Ac-225, [^225^Ac]Ac-DTPA	7.5 ± 1.0	7.3 ± 1.5	9.6 ± 2.9
Radiolyzed [^225^Ac]Ac-PSMA-I&T	1.8 ± 0.9	1.9 ± 1.0	1.5 ± 0.3
[^225^Ac]Ac-PSMA-I&T	90.2 ± 0.2	90.3 ± 0.3	89.1 ± 3.5

**Table 5 pharmaceutics-13-00715-t005:** Quality control results, for release for clinical use of [^225^Ac]Ac-PSMA-I&T injection solution.

Release Criteria	Acceptance Limit	Batch 1	Batch 2	Batch 3
RCY (Radio-(i)TLC-scanner)	>95%	97.0%	97.2%	96.3%
RCY (HPGe-detector)	>95%	99.9%	99.7%	99.7%
RCP/stability (HPLC)	>90%			
T0h		90.0%	90.1%	90.4%
T3h		90.1%	90.6%	90.2%
Bacterial endotoxins	<5 EU/mL	Conform	Conform	Conform
pH	5.5–9	5.5	5.5	5.5

## Data Availability

Not applicable.

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
