# Peer review of "Development of [225Ac]Ac-PSMA-I&T for Targeted Alpha Therapy According to GMP Guidelines for Treatment of mCRPC"

_pharmaceutics, 2021, doi:10.3390/pharmaceutics13050715_

Round 1

Reviewer 1 Report

This study presents the preparation of Ac-225 labeled PSMA, which is a promising therapeutic agent for prostate cancer, according to GMP guidelines. The targeting effectiveness of PSMA is well evaluated and thus alpha-emitter labeled PSMA will be an interesting radiopharmaceutical for the treatment of cancer cells. 

I'd suggest some revision of this manuscript. 

1) Please add the structure of PSMA, [225Ac]Ac-PSMA-I&T, and its possible decomposed (radiolysed) structure(s). In addition, provide the scheme showing radiolabeling procedure. These figures will be helpful to the readers.

2) To increase the stability during the radiolabeling, a high concentration of ascorbate was added to the reaction mixture. What is the final concentration of ascorbate in the radiolabeled PSMA? How much residual ascorbate in the final solution is allowed for a clinical trial?

3) To increase the stability, the authors used increasing amounts of PSMA ligand. That resulted in increased RCY, however will also affect the efficacy of cancer cell treatment. Please discuss. 

4) Please add detailed discussion on how to optimize the radiolabeling process in the future for the clinical trial. 

5) HPLC condition (eluent information) is not clear. 'B) acetonitrile containing 5% 0.1% TFA' means 95% acetonitrile/5% H2O containing 0.1 % TFA ?

Author Response

Thank you for your contribution. Your comments were highly appreciated. 

Reviewer 2 Report

see uploaded file

Author Response

(The authors gave the same response as above.)

Round 2

Reviewer 2 Report

While the authors addressed most of the issues, some do remain. 

  1. Although the writing has improved somewhat, there is still considerable room for improvement. An annotated manuscript is provided.
  2. Did not respond to the second part of issue 2 of the last review.
  3. Page 3, line 126: Typically sample is spotted at the origin (Rf = 0). Why was it placed at Rf = 1.0?  In Table 1, it says the activity was spotted at Rf = 0.1.
  4. Table 1: Did they mean 0.5x10 cm for the size of the iTLC strip? Need to include dose calibrator in this Table.
  5. Page 6, line 224: Shouldn’t the RCY be ratio (or %) of total activity in the product fraction rather than the ratio of product fraction to unreacted activity fraction?
  6. Page 10, Discussion, 2nd sentence: Zacherl et al did do a clinical study of this radiopharmaceutical. Although not sure whether they published it, I would think they must have done similar study.  Thus, can it be claimed that this is the first study?
  7. Page 11, line 383: What is “RCY quality control results”?

Author Response

Thank you for your constructive feedback, we sincerely appreciate your review.
